# Deformation Process of 3D Printed Structures Made from Flexible Material with Different Values of Relative Density

**DOI:** 10.3390/polym12092120

**Published:** 2020-09-17

**Authors:** Paweł Płatek, Kamil Rajkowski, Kamil Cieplak, Marcin Sarzyński, Jerzy Małachowski, Ryszard Woźniak, Jacek Janiszewski

**Affiliations:** 1Faculty of Mechatronics and Aerospace, Military University of Technology, 2 Gen. S. Kaliskiego Street, 00-908 Warsaw, Poland; kamil.rajkowski@wat.edu.pl (K.R.); kamil.cieplak@wat.edu.pl (K.C.); marcin.sarzynski@wat.edu.pl (M.S.); ryszard.wozniak@wat.edu.pl (R.W.); jacek.janiszewski@wat.edu.pl (J.J.); 2Faculty of Mechanical Engineering, Military University of Technology, 2 Gen. S. Kaliskiego Street, 00-908 Warsaw, Poland; jerzy.malachowski@wat.edu.pl

**Keywords:** 3D printing, Fused Filament Fabrication, thermoplastic polyurethane, energy absorption, dynamic compression, crashworthiness, Simplified Rubber Material, Ls Dyna

## Abstract

The main aim of this article is the analysis of the deformation process of regular cell structures under quasi-static load conditions. The methodology used in the presented investigations included a manufacturability study, strength tests of the base material as well as experimental and numerical compression tests of developed regular cellular structures. A regular honeycomb and four variants with gradually changing topologies of different relative density values have been successfully designed and produced in the TPU-Polyflex flexible thermoplastic polyurethane material using the Fused Filament Fabrication (FFF) 3D printing technique. Based on the results of performed technological studies, the most productive and accurate 3D printing parameters for the thermoplastic polyurethane filament were defined. It has been found that the 3D printed Polyflex material is characterised by a very high flexibility (elongation up to 380%) and a non-linear stress-strain relationship. A detailed analysis of the compression process of the structure specimens revealed that buckling and bending were the main mechanisms responsible for the deformation of developed structures. The Finite Element (FE) method and Ls Dyna software were used to conduct computer simulations reflecting the mechanical response of the structural specimens subjected to a quasi-static compression load. The hyperelastic properties of the TPU material were described with the Simplified Rubber Material (SRM) constitutive model. The proposed FE models, as well as assumed initial boundary conditions, were successfully validated. The results obtained from computer simulations agreed well with the data from the experimental compression tests. A linear relationship was found between the relative density and the maximum strain energy value.

## 1. Introduction

Over the last two decades, a growing development of additive manufacturing (AM) techniques has been observed [1,2,3,4]. The diversity of techniques available and a wide range of materials used with different physical properties make this type of manufacturing methods very attractive to many industrial sectors [5]. In addition, the design freedom offered by AM allows for manufacturing of objects with complex shapes that are otherwise not possible with conventional subtractive production methods [6,7]. Among the most popular AM techniques offered in the market are Fused Deposition Modelling (FDM) and Fused Filament Fabrication (FFF), which came into being after the Stratasys Corp. (Eden Prairie, MN, USA) patents expired. The two techniques are commonly referred to as 3D printing [8,9,10]. This name comes from the technical solution used during the object building process, where the thermoplastic material (filament) is melted and then extruded through a nozzle to build up the layers that gradually make up a complete part. The FDM technology marketed by Stratasys Corp is generally intended for professional applications and uses a limited number of materials. However, their quality and mechanical properties are certified by the producer. Fused Filament Fabrication 3D printers offered in the market by many manufacturers have generally become very popular due to low prices and wide range of materials available. Acrylonitrile-butadiene-styrene-ABS, polylactic acid-PLA, and polyamide are among the most popular and most frequently used [11,12,13,14,15]. In general, this is caused by optimised parameters of the technological process and widely discussed methods of reducing the negative effects of thermal shrinkage. In recent years, an increasing number of special materials (flexible, conductive, magnetic, ferromagnetic, with shape-memory) dedicated to the FFF 3D printing technique have been observed [16,17,18,19]. They come as a response to new technological needs. Some of them have multifunctional properties that offer the possibility of using them in cutting-edge products in the aerospace industry [11], the automotive industry, robotics [14], electronics, and biotechnology [20,21]. Reinforced with fibres, they exhibit a high mechanical strength, significantly higher than typical ABS or PLA materials [22,23,24]. Some of the representatives of the mentioned group of special filaments are flexible and high plasticity materials [17,25]. They demonstrate a high range of deformation under various loading conditions, where the damage mechanism is usually caused by buckling, bending, and shearing [26]. These properties are particularly important for energy absorption and crashworthiness applications, where a wide range of material deformations without a damage mechanism such as cracking suddenly occurring are required [27,28,29,30,31,32].

Based on the literature study conducted, it was found that thermoplastic materials are widely used in the process of manufacturing regular cell structures, demonstrating high efficiency in terms of energy absorption [33,34,35]. The advantages of the additive structure manufacturing make it possible to tailor the deformation process of the structural specimen according to the applied material and elementary singular cell topology [36,37,38,39]. Many available papers present results obtained for 2D structural specimens with different topologies made from ABS or PLA [40,41]. Hedayati et al. [34] presented a mechanical response of 3D printed honeycomb specimens made from PLA under quasi-static in-plane compression tests. They analysed the influence of the specimen wall thickness value on the deformation process. Kucewicz et al. [42] studied the behaviour of various topologies of ABS structures under quasi-static and impact loading conditions. Furthermore, they developed FE models taking into account the material damage mechanism, which resulted in a good agreement with the experimental studies [43]. Yang et al. [44] carried out tensile and compression tests on specimens, taking auxetic topologies into account. In addition, they analysed the mechanical response of a structure with gradually changing elementary singular cell topology to the deformation process. Chang et al. [45] focused on the in-plane crushing response of tetra-chiral (TC) honeycombs exhibiting auxetic properties. They carried out experimental and numerical investigations taking into account the impact load and quasi-static conditions. The main conclusion of their study is that the relative density has a strong influence on the in-plane crush strength value of the sample. Tabacu and Ducu in [46] presented the results of their research where they evaluated multicellular ABS inserts subjected to compression tests. Using an experimental and numerical approach, they discovered that ABS, multicellular inserts with an outer aluminium tube, designed by them, gave a progressive profile of crushing force during out-plane compression. The application of the proposed solution allows to significantly increase the level of protection of thin-walled structures used in transport systems. Bodaghi et al. [47] have conducted interesting studies on soft PLA material manufactured additively with the use of FFF. They revealed that the singular cell shape, direction, type, and magnitude of mechanical load affects the metamaterial’s anisotropic response and its instability characteristics.

With reference to the work mentioned, it can be observed that additive manufacturing attracts the attention of many scientists in the field of crashworthiness and energy absorption studies. However, only a few of them were made with the use of flexible 3D printing materials. This is generally due to the fact that the 3D printing process with flexible material is more demanding and cannot be realized with all FFF 3D printers available in the market. Furthermore, the determination of suitable technological parameters requires additional, time-consuming optimisation studies.

The main aim of this paper is to present results regarding the mechanical response of 3D printed regular cellular structures made from thermoplastic polyurethane material (TPU 95 Polyflex, sold by Polymaker Corp. Shanghai, China), subjected to quasi-static in-plane compression tests. The main feature of the applied material is high flexibility that results in a high deformation range. This property, combined with the correct topology of the regular cell structure, can lead to high efficiency in energy absorption and crashworthiness applications. For this reason, the authors decided to conduct studies in which the honeycomb topology and its modification were used to analyse the deformation plot of the structure under quasi-static load conditions. The proposed research consists of two parts. The first is related to experimental tests, in which few variants of honeycomb topologies (regular and gradually changing) with various relative density values were manufactured using the 3D printing technique and then subjected to in-plane compression tests. The other part of the research involved computer simulations performed using finite element analysis. On the basis of the collected results, conclusions were formulated on the influence of the proposed topologies on the deformation process of the structure. The research carried out by the authors as a result could make it possible to find an effective energy absorption solution that could be applied in military protection systems, as well as in automotive, aerospace, electronics, and biotechnology applications. The main steps of the applied research methodology used to conduct the studies are shown in Figure 1.

The first step was related to the development and manufacture of structural specimens with different relative density values. The next task was the geometric quality control of fabricated structure specimens. Then, the characterisation of the mechanical properties of the applied TPU 95 Polyflex filament was performed. The last two stages were related to investigations of the deformation process of the structural specimens. The former stage was based on experimental compression tests, while the latter stage was based on the application of FE analysis and Ls-Dyna software. Based on the results obtained, the influence of the structure topology on its deformation history plot was discussed.

## 2. Development and Fabrication of Structural Specimens

The main idea considered during the structural specimen design process involved the application of typical honeycomb topology (HC) as the reference model. In Figure 2, other topology variants were defined as gradient models developed based on the proposed geometric features of honeycomb specimen. The main geometric assumptions made during this phase are the following: The total size of the specimen must be less than 40 mm, with a minimum of seven cells in a single layer of the structure. The total specimen’s dimension value of less than 40 mm is the result of the diameter of the Split Hopkinson Pressure bar laboratory stand that is planned to be used for further dynamic compression tests. Taking into account guidelines regarding the investigation of regular cell structures, it is recommended to use a minimum of seven cells in a single structure layer to properly evaluate the influence of topology on the specimen deformation history plot. For this reason, the specimens were defined with the same overall dimensions (height-31.4 mm, width-34.4 mm, depth-20 mm), but with different dimensions of the elementary singular cells. As a result, it was possible to obtain different values of the structure relative density *ρ*_rel_.

Detailed information on the dimensions of the developed structural specimens has been presented in Figure 3 and Table 1. Specimens No.2 and No.3 are unidirectional, linear gradients of the honeycomb topology (HCG). The main difference between them is the direction of decrease (HCG_D) or increase (HCG_D) of the value of the elementary singular cell size. Specimen No.4 consists of two elementary singular cells that enable the definition of the topology with discrete gradients (HCG_Ds). The size of the particular singular cells was chosen so that the global dimensions of the specimen would be equal to the reference HC model. Specimen No.5 was characterised by a singular cell size that gradually changed in two directions (HCG_BD). The nominal wall thickness dimension of the specimen (1.0 mm) was established based on the results of preliminary technological studies in which a flexible material was used.

The next stage of the studies concerned the process of manufacturing the structural specimens. Based on preliminary test results, it was found that the flexible material (e.g., thermoplastic polyurethane) indicates prospective mechanical properties that could be used in energy absorption applications. Studies conducted using typical materials used in the FFF 3D printing techniques (ABS, PLA) showed a tendency for rapid cracking due to a low range of plastic deformation. Taking this feature into account, the authors decided to conduct a preliminary investigation with flexible materials, which are more demanding in terms of defining the appropriate parameters of the technological process but give a greater range of deformation. Considering the flexible materials available on the market, it was decided to use the thermoplastic polyurethane TPU 95-Polyflex distributed by Polymaker Corp. This material has a Shore hardness of 95 A and can elongate more than three times its original length. The mechanical properties of the material used provided by a producer are presented in Table 2.

The high flexibility of the proposed materials affects their manufacturability and limits the number of FFF 3D printers that can be used to conduct the manufacturing process. In general, devices with a direct filament feed mechanism are dedicated to carrying out this process. For 3D printers with a Bowden feed mechanism, where the distance between the feed mechanism and the end of the extruder is relatively greater, the use of a highly flexible filament is not recommended. The FFF 3D printer used to fabricate the developed structure specimens and samples used for the characterisation of materials is presented in Figure 4. It was a Prime 3D, a low-cost device manufactured by the Polish Monkeyfab Corp (Warsaw, Poland). However, the authors made some modifications to the extruder construction to improve the heat flux distribution. The hot-end block was made from an aluminium alloy and additional cooling fins were added to increase the cooling efficiency of the extruder tip. In addition, an extra fan was installed on the back of the extruder system with a directional airflow housing system. Due to this modification, the presented 3D printer allowed for manufacturing process with the use of a wide range of thermoplastic and flexible materials.

Rubber-like filaments are flexible and require time-consuming studies to find the most efficient 3D printing parameters that ensure high structural and geometric quality of manufactured objects. For this reason, the specimens manufacturing process was preceded by additional technological studies. They focused on optimising the 3D printing process when using the TPU 95 Polyflex filament. Additional benchmark model presented in Figure 5a has been designed in a SolidWorks CAD system to perform these tests. It consists of a series of single walls, square tubes, and hexagonal singular cells. All elements were designed with different wall thicknesses. By changing the values of selected 3D printing parameters, such as material flow, nozzle and table temperatures, layer height, and a fill pattern, the quality of manufactured objects was evaluated. The main criteria that were considered during these studies were: A maximum ratio of filling of the model by the material, absence of voids and imperfections in the material, regular shape and height of a single layer, and minimal deviations in wall thickness. The geometrical measurements (the results are presented in Figure 5b,c) were carried out with the use of VI-2510 Venture (Baty Corp. Albany Court, UK) Optical Coordinate Measuring Machine (CMM).

The influence of the 3D printing parameters on the quality of the manufactured model can be discussed using the data presented in Table 3 and Table 4. Table 3 contains sample sets of 3D printing parameter groups that were changed during tests. Table 4 presents the measured values of the characteristic dimensions (wall thickness) defined in Figure 4. From these data it can be seen that a very important parameter is the material flow ratio that affects the filling of a model. However, excessive material flow causes a significant dimensional deviation in the measured wall thickness. Based on the results of many tests carried out, it was found that in the case of the used 3D printer as well as the TPU 95 material, the most effective group of 3D printing parameters is the set number 3 presented in Table 3. This group of parameters was used in further fabrication processes of samples used in material characterisation, as well as structural specimens used during the experimental tests.

## 3. Structural and Geometrical Quality Control

The next stage of the study carried out was related to structural and geometric quality control. The Keyence VHX-600 digital microscope (Keyence, Osaka, Japan) was used to evaluate the structural quality of fabricated specimens. Based on the photos taken, it was determined that the parameters of the 3D printing process were properly adopted. By looking at the upper surface of the specimen, a uniform way of filling the specimen structure with material can be seen (Figure 6a). The wall thickness was determined by three paths of the working nozzle. There are no visible voids or other imperfection of material filling (Figure 6b,c). In addition, Figure 7 shows the side view of the specimen. All filament layers are set correctly, they show a similar height, and no significant irregularities were observed after the manufacturing process.

The other important issue evaluated during the quality control process was the ratio of wall filling by material, and verification of internal imperfections that could affect the mechanical response of the structure during compression tests. This evaluation was made based on the digital microscopic observation of the sectional views presented in Figure 8 and Figure 9. These observations revealed the presence of small voids in the mid-section of the wall thickness of the sample. They have a regular character and were caused by the method of building up the wall thickness (three paths without additional filling). However, the low single-layer height and the high elasticity of the TPU filament material mean that the influence of the observed imperfection on the mechanical response of the structure under compression tests is rather negligible when compared to other materials, like ABS or PLA.

Another step in the quality control process involved the determination of the dimensional deviations of the specimens. This problem is particularly critical in terms of defining accurate FE models of structures required to conduct computer simulations. Wall thickness measurements of the structure were performed using the Keyence VHX-600 digital microscope. Images were taken with a magnitude of ×20. Figure 10 shows the results of the measurements obtained for the honeycomb as a representative topology.

Despite the preliminary manufacturability tests performed to determine the correct parameters of the 3D printing process, a high geometric deviation of the fabricated wall thickness of the cellular samples can be observed. Based on carried out measurements with the use of digital microscopy, the estimated average value of structure wall thickness was 1.1 ± 0.065 mm. Measurements were performed based on three various specimens of regular honeycomb topology. This situation can be explained by a high material flow during the extrusion of filaments (controlled by the temperature value of the nozzle and the speed of the material feeding mechanism) which had to be used to guarantee a high structural quality. Observed discrepancies between nominal and real values of specimen wall thickness were included in the developed FE models that were used in the numerical simulations.

## 4. Determination of Mechanical Properties of the TPU Material

The further step of the investigation was related to the determination of the mechanical properties of the applied TPU 95 Polyflex material. Both the uniaxial compression and tensile tests were carried out in accordance to the PN-EN ISO 604:2006 and PN-EN ISO 37:2011 standards under normal temperature conditions. The shape and the dimensions of the samples used during the tests are shown in Figure 11. The tests were carried out using an MTS Criterion C45 universal strength machine. The loading velocity applied in both tests was 1.0 mm/s.

The results obtained from the tensile tests are presented in Figure 12. Taking into consideration the repeatability of registered test data, carried out at room temperature, it has been decided that the number of attempts could be limited to three for compression and three for tensile tests. When analysing the history plot, the non-linear, hyper-elastic mechanical properties of TPU 95 Polyflex filament material could be observed. In addition, it showed a high deformation range during tensile tests. The tensile elongation of specimens was approximately 380 ± 0.36% and the maximum stress value was approximately 38.3 ± 0.21 MPa. These values differ from the data for the TPU 95 Polyflex filament. This discrepancy can be explained by the technological process parameters adopted during the manufacturing process, such as: The height of the layer and the type of specimen filling pattern applied. The obtained material characterisation results were used in subsequent research related to computer simulations.

## 5. Experimental Investigations of Mechanical Properties of the Structural Specimens

The next stage of the investigations carried out focused on defining of the mechanical response of the proposed structure topologies under quasi-static load conditions. Compression tests were performed using the MTS Criterion C45 universal strength machine. The use of a digital camera made it possible to register the deformation process step by step. The structure orientation in relation to the direction of the load applied was the same in all tests. As a result, deformation history plots presented in Figure 13, Figure 14, Figure 15, Figure 16 and Figure 17 were defined. Furthermore, thanks to the recorded videos it was possible to analyse in detail the mechanical behaviour of the structures during the compression tests. Taking into account the high repeatability of registered data, it was decided that the number of specimens used in these tests could be limited to three for particular structure topologies. By referring to the images and plots that recorded the mechanical response of the reference honeycomb (HC) topology (Figure 13), three main stages of deformation can be highlighted. The first stage (marked between points from No.1 to No.2) refers to a linear deformation resulting from the mechanical properties of the applied material and geometric stiffness of the singular cells. The next stage of deformation (highlighted between points from No.2 to No.3) is caused by the loss of stability and the arrival of buckling and bending mechanisms in one of the structural layers. A further deformation stage is reflected in the plot as a long plateau (marked between points No.3 and No.4). The final stage, represented by the rapid growth in the value of the deformation force, is linked to the collapse of all arrays inside the specimen and the final densification of the structure (marked between points from No. 4 to No.5).

Subsequent variations of the specimens subjected to compression tests were the gradual honeycomb topologies in which singular cell size changed linearly (decrease-HCG_D and increase-HCG_I). The two topologies show a similar relative density value (*ρ*_rel_-0.39), which is greater than the representative honeycomb topology’s value (*ρ*_rel_-0.37). When analysing the stages of their deformation process (Figure 14 and Figure 15), three main stages can be distinguished. However, the range of the plateau is relatively small compared to the honeycomb. The first stage, similar to the previous case, is linear and results from the mechanical properties of the applied TPU 95 material and geometric stiffness of the specimen (marked on the curve by points from No.1 to No.2). However, a larger number of singular cells that are in the direction of the acting load leads to a higher value of the registered deformation force. The next stage of deformation is related to the loss of structural stability caused by the buckling and bending mechanisms and the decrease in the deformation force value (highlighted on the curve by points from No.2 to No.3). Taking into account the large difference in stiffness between the largest singular cells and the rest, the array with the largest cells collapsed first, and this way the process of progressive structure deformation has been initiated (marked by points from No.3 to No.4). This process is caused by buckling and bending mechanisms, that arrived progressively in particular layers of the structure. The last stage marked on the curve by points from No.4 to No.5 is related to the deformation of the remaining arrays followed by the final densification. When comparing the deformation process of the HCG_D and HCG_I specimens, it can be determined that they show a similar history plot. However, due to the different locations of the arrays consisting of the larger singular cells, they collapsed differently—the structure HCG_D collapsed starting from the inside, and the HCG_I–starting from the outside.

The following results in Figure 16 refer to the honeycomb topology with discrete gradients (HCG_Ds). It consists of two types of singular cells that differ in size. When comparing the obtained results with the plots registered for previously discussed specimens, it can be noticed that the deformation force history plot is similar to the regular honeycomb topology. The range of the first, linear stage of the deformation process (marked on the curve by points from No.1 to No.2) results in a higher value of the deformation force, which is generally due to a higher geometric stiffness of the topology of a singular cell. The next stage was related to the loss of structural stability caused initially because of buckling and then bending mechanisms. This period of deformation begun at point No.2 and lasted until point No.3. However, the plateau range is shorter due to a limited number of larger cells located in the direction of the loading action (marked on the curve by points from No.3 to No.4). Furthermore, the buckling effect of the whole structure was observed in the final stage of the plateau before densification which begun in point No.4 and lasted until point No.5.

The last of the results obtained during the compression tests refer to the bidirectional gradual honeycomb topology (HCG_BD) presented in Figure 17. Due to a higher value of the relative density (*ρ*_rel_-0.42), the stiffness of the structure was relatively high compared to the representative regular honeycomb. The value of the deformation force registered in the first linear stage of the deformation process (marked on the curve by points from No.1. to No.2) was almost two times higher. After the loss of the structural stability, caused by buckling and bending mechanisms, the gradual decrease of deformation force was registered (highlighted by points from No.2 to No.3). This stage lasted until the collapse of the layers defined by cells with the largest size. Afterward, the progressive deformation of the whole structure could be observed (from point No.3 to point No.4). The final densification happened faster compared to other structural topologies due to a higher relative density value (marked on the curve from point No.4 to point No.5).

Based on the results of performed compression tests, it can be stated that the main mechanisms responsible for the deformation of the particular structures were buckling and bending. Taking into account the high flexibility of the applied base material, there was no presence of shearing or cracking mechanisms.

Results of experimental compression tests made it possible to compare the honeycomb-based topologies proposed by the authors in terms of energy absorption efficiency. The estimated values of the deformation energy are shown in Figure 18. The influence of the proposed topology on the deformation energy value as well as on the deformation range can be observed. In fact, it is interesting that the use of a gradually changing topology with a similar relative density value with respect to the reference honeycomb structure made it possible to obtain a higher value of the deformation energy. The same conclusion can be formulated from the data presented in Figure 19, where the value of the absorbed energy was related to the structural volume (*SEA_v_*), defined by the Formula (1):(1)$$SEA_{v}=\frac{EA}{V_{s}}$$
where *EA* is the energy absorption and vs. is the volume of the structure specimen

Furthermore, the application of the TPU 95 Polyflex material, which exhibits hyper-elastic mechanical properties, made it possible to reduce the negative effects of the material cracking during the deformation process. After unloading, their dimensions were similar to those from before the tests.

## 6. Numerical Investigations of Mechanical Properties of Structural Specimens

The experimental compression tests conducted have allowed to draw a conclusion about the possibility of using gradient honeycomb topologies and the TPU 95 Polyflex materials as an interesting solution in terms of energy absorption. However, the experimental approach in further studies to optimise the structure topology is costly and time-consuming. For this reason, it was decided to use an additional numerical modelling approach based on the Finite Element Analysis (FEA). Computer simulations were conducted with the commercial Ls-Dyna hydrocode with Multi Parallel Processing (MPP) (Livermore Software Technology, Livermore, CA, USA) [48]. The main assumptions made during computer simulations were similar to those presented by the authors in [42,49,50]. The FE models required to run the simulation of the particular structural behaviour under the compression tests have been developed from the CAD models (Figure 20). Eight-node hexagonal elements with single integration point were used to create FE models. They were defined as a 2 mm sections in all structures. Based on the results of previous studies [43,49], this assumption allowed a significant reduction in simulation time and caused no impact on the final result’s accuracy. Taking into account the geometric deviations of the manufactured specimens observed during the geometrical quality control, the wall thickness of the specimen was defined as 1.10 mm and consisted of four elements.

Detailed information on the type of computer simulation is shown in Figure 21. The applied boundary conditions corresponded to the quasi-static tests. The structural specimen was placed between two rigid planes. The bottom plane was fixed with the top plane moving in the 0Y direction. The plane motion *v*(*t*) was prescribed by the Formula (2) used in previous papers [43,51].
(2)$$v(t)=\frac{\pi }{\pi -2}\frac{S_{\max}}{T}\left[1-\cos\left(\frac{\pi }{2T}\right)\right]$$
where *T* is the termination time of the simulation and *S*_max_ is the final displacement of the rigid surface.

A Single-Surface and Surface-to-Surface contact settings based on the penalty method were used to simulate the interactions between the planes and the structure, and the self-interaction of the structure cells. A Coulomb formulation was used to describe the tangential interaction between bodies [48] with the friction coefficient *µ*-0.65 for the TPU 95 Polyflex material.

Ls-Dyna software allows for use of a few constitutive material models based on the Ogden formulation, which are suitable for describing hyperelastic materials. Nevertheless, based on the conclusions presented in the following articles [52,53,54,55], the authors decided to use the Simplified Rubber Material (SRM) model. It is defined on the basis of the experimentally determined tabulated stress-strain curve (Table 5). Furthermore, the SRM material model allows the effects of strain rate to be taken into account based on the series of stress–strain curves that have been experimentally defined under different load conditions. This feature is intended to be used in further studies related to dynamic compression tests. In addition, Table 6 presents the material parameters used to define the SRM.

The results obtained during the computer simulations were compared with the data recorded during the experimental tests to verify the accuracy of the initial boundary conditions adopted as well as the applied constitutive model describing the mechanical properties of the TPU 95 Polyflex. In Figure 22, the history of the deformation forces of the honeycomb topology is shown. The numerical results obtained, which are highlighted by a black dotted line, corresponds to the data recorded during the experimental tests. In addition, the same mechanisms, such as buckling and bending, which are responsible for the deformation process of the specimen, were observed. Applied material model without definition of failure criteria resulted in lack of shearing and cracking effects.

The following results refer to the honeycomb gradual topologies. In Figure 23 and Figure 24, the results that reproduce the deformations of HCG_D and HCG_I are presented. Both agree well with the experimental data. The estimated maximum values of deformation forces at each compression stage are similar to each other. The main difference between the used topologies is the location of the initialisation of the specimen collapse process. In the case of the HCG_D topology, this process started in the inner layer of the specimen, while in the case of the HCG_I—in the outer layers. In the analysis of the main mechanisms responsible for the specimens deformation process, similar to regular honeycomb topology, the buckling and bending mechanisms were observed. Similar to experimental tests, there are no shearing or cracking effects.

The subsequent results shown in Figure 25 correspond to the honeycomb topology with discrete gradients (HCG_Ds). The first linear compression stage agrees well with the experimental data. The visible difference in the deformation force value in the plateau range was likely caused by the initially assumed wall thickness of the smaller singular cells of the FE model. In reference to the observations made during the technological studies, the adopted 3D printing technique as well as the TPU 95 filament requires individual 3D printing parameters depending on the type of objects printed. A large number of geometric elements of small dimensions require different 3D printing parameters. Generally, in this case, it is recommended to reduce the rotation velocity of the stepper motors in order to reduce the material flow. By using the same parameters for the HCG_Ds topology as for the referential HC, the dimensional deviations of the small singular cell thickness for HCG_Ds are greater. This situation explains the difference between the numerical results and the experimental data. Referring to the main mechanisms arriving during structure compression tests, similar to experimental results, only buckling and bending mechanisms could be observed.

The last considered topology refers to the bidirectional modification of the honeycomb gradient. Figure 26 shows the results obtained from the FE analysis. When comparing the quality of the proposed numerical model and the assumed initial boundary conditions, a conclusion similar to that of the of HCG_D topology variants can be drawn. The initial linear phase of the deformation has been reproduced accurately, however there is a visible difference in the deformation force history plot in a subsequent stage where some of the layers of the structure have collapsed. These differences are mainly due to the dimensional deviations of the small singular cells the topology of the structure consists of. Referring to the main mechanisms responsible for specimens collapse, the buckling and bending mechanisms can be distinguished.

Numerical investigations carried out have made it possible to predict the mechanical behaviour of the topologies proposed during quasi-static compression tests. The presented results of deformation history plots generally are in a good agreement with the data obtained during the experimental tests. In the case of gradient topologies, however, certain differences could be observed in the history plots obtained. This situation was mainly caused by dimensional deviations of actual structure specimens where small-sized singular cells were used. This characteristic can also be observed in Figure 27, where the comparison of deformation energy history plots was presented. The solution that can reduce this discrepancy requires an individual evaluation during the particular FE model design process.

## 7. Conclusions

Based on the research conducted regarding the compression tests of regular and gradual honeycomb structure specimens, manufactured additively from flexible material, the following conclusions were made:-Fused filament fabrication is a cost-effective 3D printing process that allows objects to be fabricated using specific filaments such as thermoplastic polyurethane (TPU 95) with unique mechanical properties. Nevertheless, the material proposed by the authors requires a direct-type filament feeding mechanism and it is recommended to conduct further technological studies in order to define the appropriate 3D printing parameters, which guarantee a structural and geometric quality of the manufactured objects.-The conducted characterisation of the mechanical properties of the TPU 95 material allows to confirm that it exhibits hyperelastic properties with a high deformation range. For this reason, it is necessary to use a suitable constitutive material model in computer simulations.-The experimental compression tests have shown a linear relationship between the relative density of the applied topology and the deformation energy value. Specimens with gradually changing topologies showed a higher value of the deformation energy compared to the reference honeycomb structure.-By analysing the history plots of the deformation processes of the specimens, a lack of crack damage mechanism can be observed due to the high flexibility of the applied TPU 95 filament. The main mechanisms occurring during the compression test were buckling and bending.-The proposed numerical approach to the investigations made it possible to predict the structural deformation process. The results obtained agree well with the data recorded during the experimental tests.-The adopted simplified rubber material constitutive model, defined on the basis of experimental compression and tensile tests, enables correct reproduction of the mechanical response of the structural specimens made with the TPU 95 filament.-The developed models will be used in further numerical investigations conducted under dynamic load conditions.-Planned investigations taking into account dynamic loading conditions will be carried out experimentally as well as numerically. The experimental approach will be performed with the Split Hopkinson Pressure Bar stand implementation in a direct impact configuration. The numerical approach will allow for verification of the proposed numerical model. A good correlation between results enables us to perform further optimisation studies.-Obtained results of the dynamic tests offer the chance for evaluation of developed structures as well as highly flexible Polyflex TPU 95a material in terms of energy absorption.

## Figures and Tables

**Figure 1 polymers-12-02120-f001:**
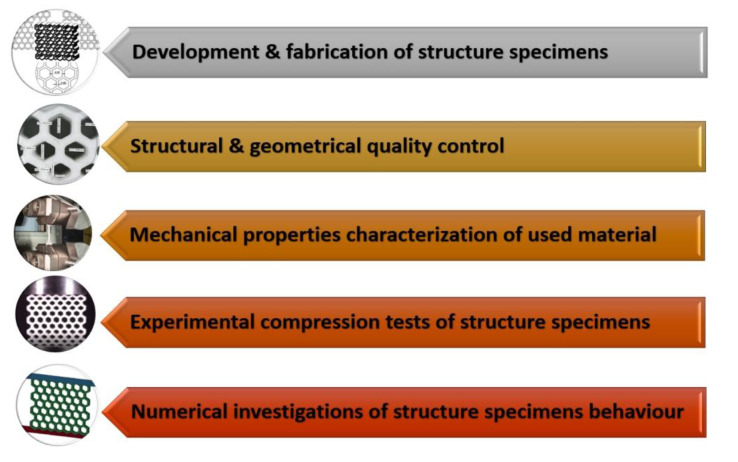
The main steps of applied research methodology.

**Figure 2 polymers-12-02120-f002:**
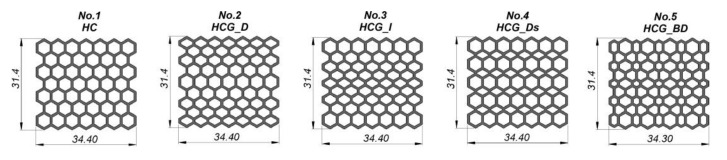
The main view of developed structure specimens with different topologies.

**Figure 3 polymers-12-02120-f003:**
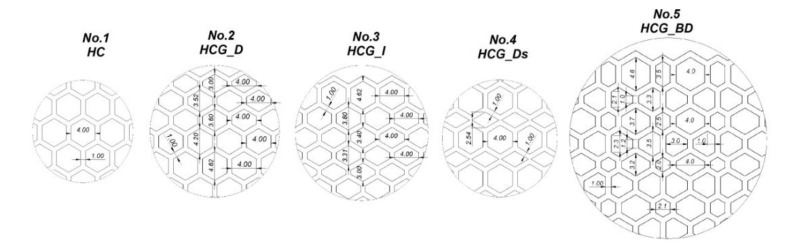
The detailed view of developed structure specimens with different topologies.

**Figure 4 polymers-12-02120-f004:**
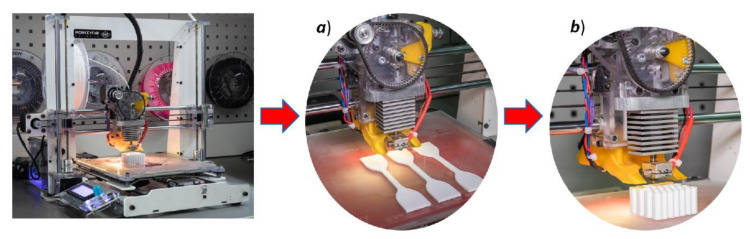
The general view of used 3D printer-Prime 3D (MonkeyFab) during the fabrication process of: (**a**) Dog bone samples, (**b**) structure specimens.

**Figure 5 polymers-12-02120-f005:**
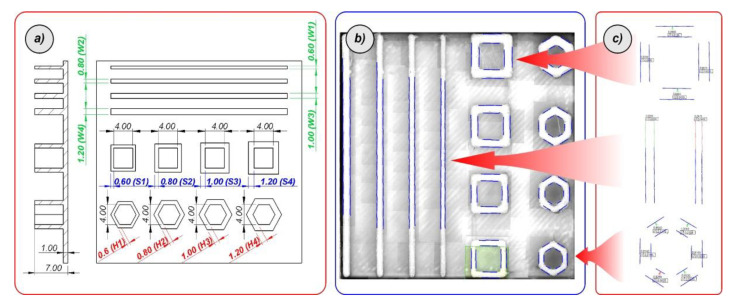
The process of optimisation the 3D printing parameters: (**a**) Developed benchmark model, (**b**) conducted measurements, (**c**) defined report with dimensional deviations.

**Figure 6 polymers-12-02120-f006:**
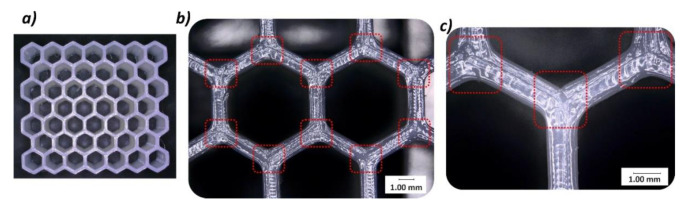
The view of the top surface of 3D printed honeycomb specimen with the use of thermoplastic polyurethane (TPU) material: (**a**) General view, (**b**) with magnitude ×10, (**c**) with magnitude ×20.

**Figure 7 polymers-12-02120-f007:**
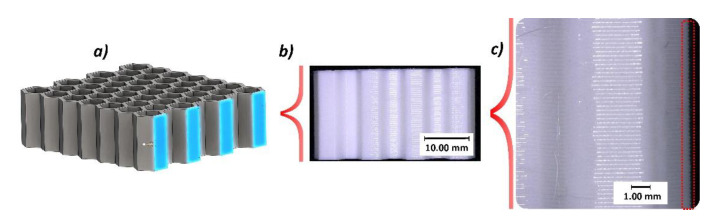
The view of the side surface of 3D printed honeycomb specimen: (**a**) 3D CAD model, (**b**) general view of specimen, (**c**) view with magnitude ×20.

**Figure 8 polymers-12-02120-f008:**
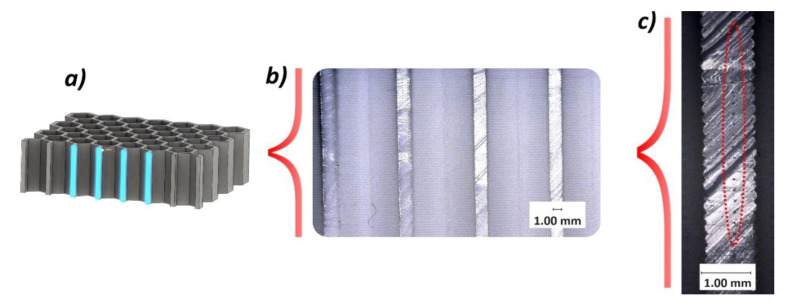
The normal sectional plane view of 3D printed honeycomb specimen: (**a**) 3D CAD model with sectional plane, (**b**) general view of specimen, (**c**) view with magnitude ×20.

**Figure 9 polymers-12-02120-f009:**
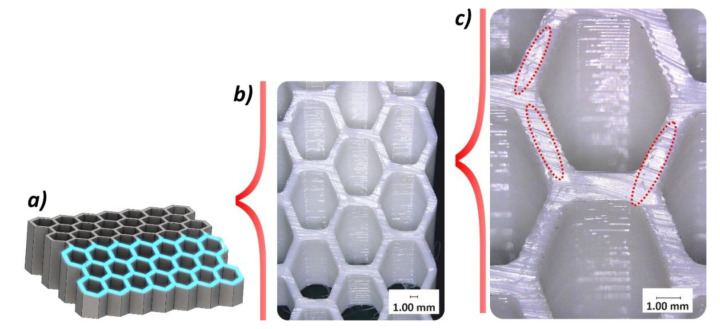
The angle sectional plane view of 3D printed honeycomb specimen: (**a**) 3D CAD model with sectional plane, (**b**) general view of specimen, (**c**) view with magnitude ×20.

**Figure 10 polymers-12-02120-f010:**
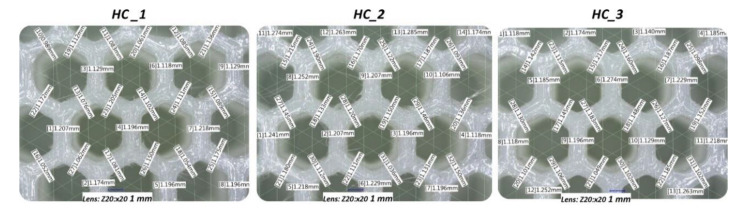
The results of honeycomb topology wall thickness measurements.

**Figure 11 polymers-12-02120-f011:**
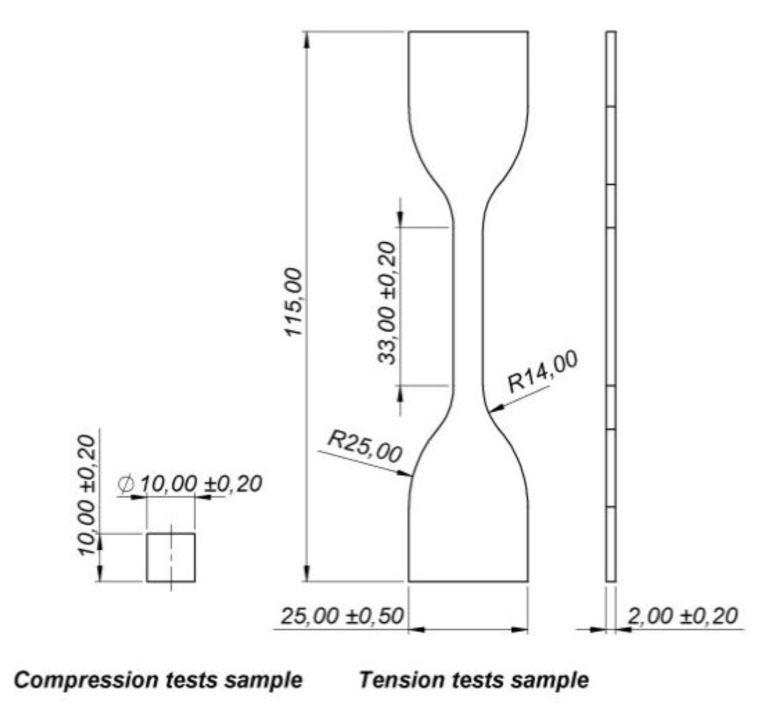
The view of the samples used during the determination of the mechanical properties of the TPU material under uniaxial compression and tensile tests.

**Figure 12 polymers-12-02120-f012:**
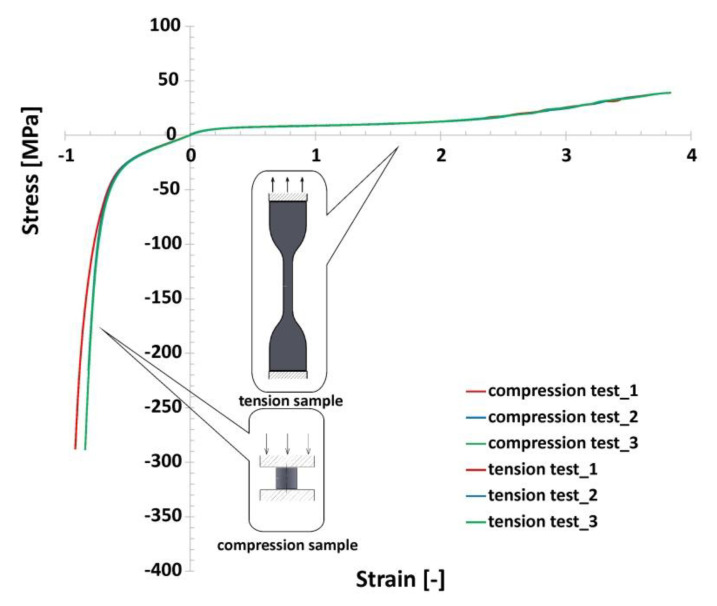
The uniaxial compression and tensile tests of the 3D printed TPU material under normal temperature conditions.

**Figure 13 polymers-12-02120-f013:**
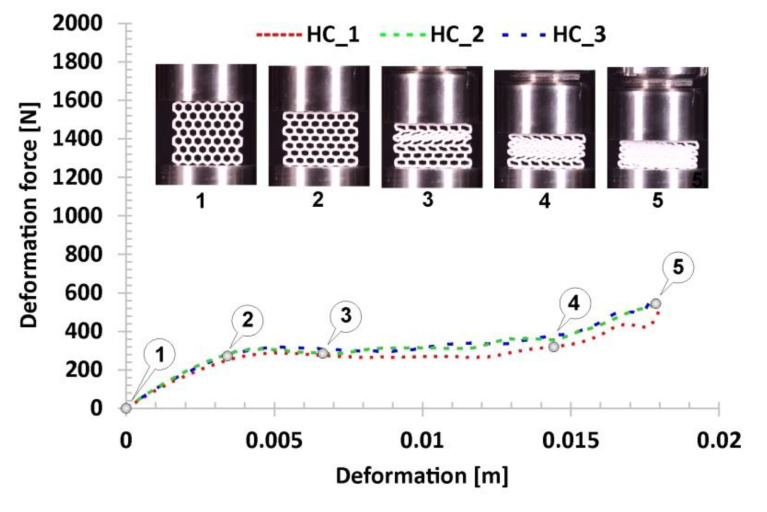
The results of the compression tests of the structural specimen with regular honeycomb topology.

**Figure 14 polymers-12-02120-f014:**
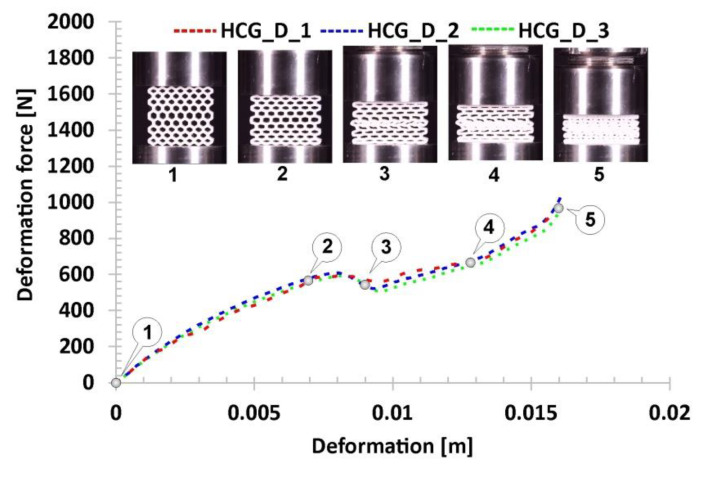
The results of the compression tests of the structural specimen with gradually decreasing honeycomb topology.

**Figure 15 polymers-12-02120-f015:**
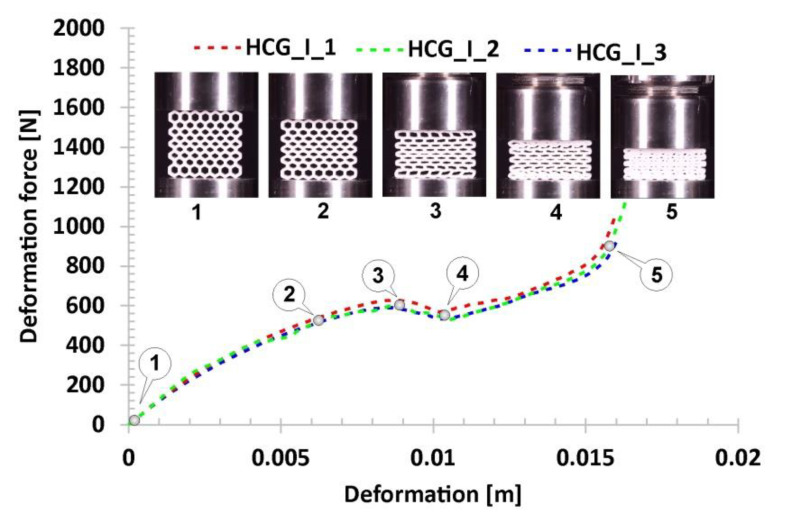
The results of the compression tests of the structural specimen with gradually increasing honeycomb topology.

**Figure 16 polymers-12-02120-f016:**
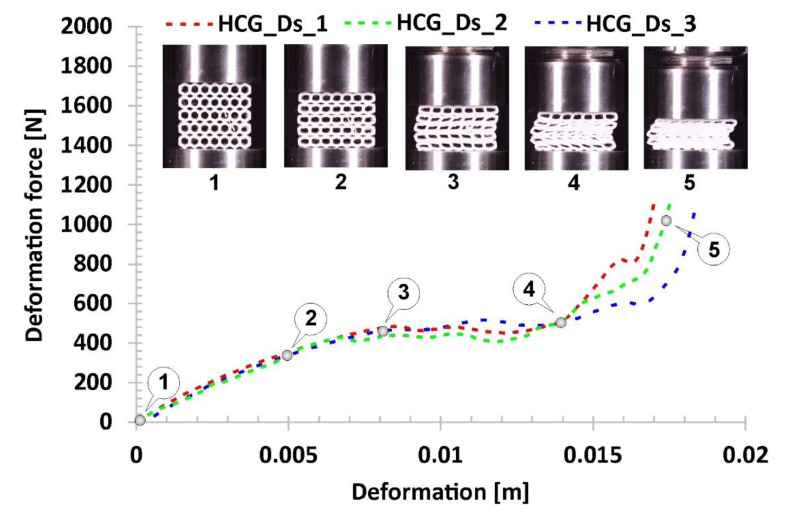
The results of the compression tests of the structural specimen with gradual discrete honeycomb topology.

**Figure 17 polymers-12-02120-f017:**
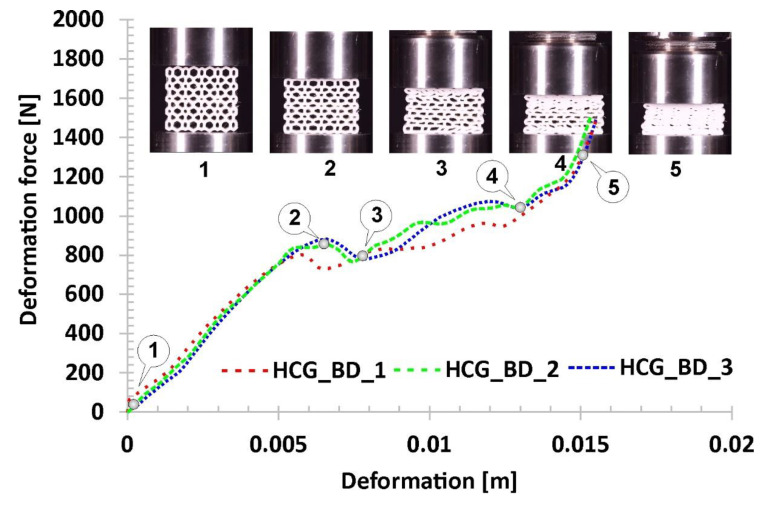
The results of the compression tests of the structural specimen with bidirectional gradual honeycomb topology.

**Figure 18 polymers-12-02120-f018:**
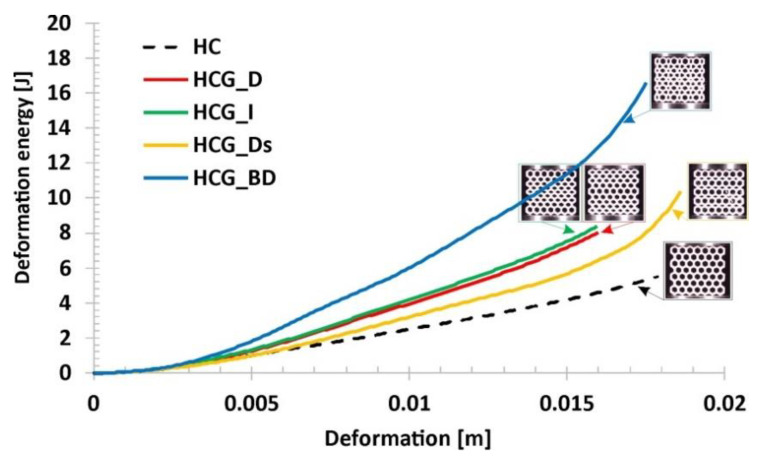
The comparison of the estimated values of the deformation energy versus the range of deformation.

**Figure 19 polymers-12-02120-f019:**
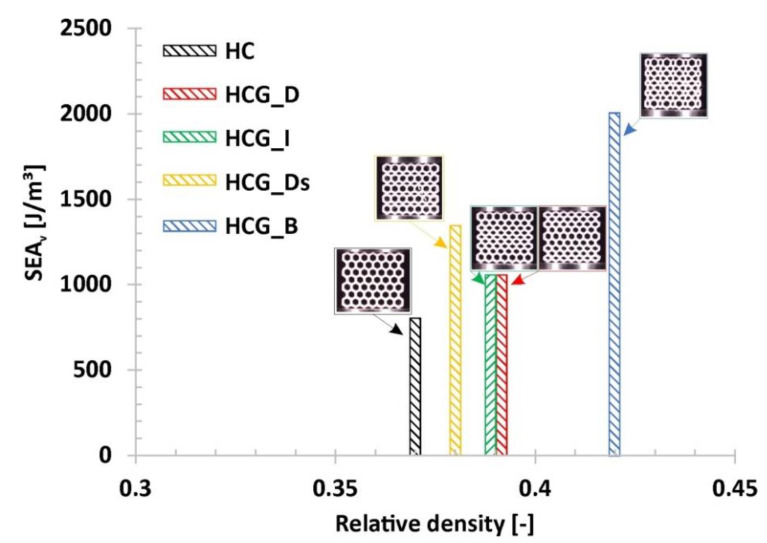
The comparison of the estimated values of the deformation energy in relation to the relative density.

**Figure 20 polymers-12-02120-f020:**
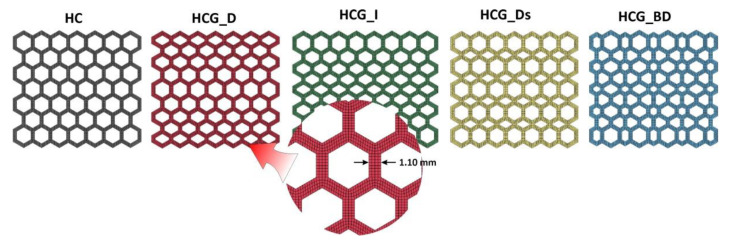
The main view of Finite Element (FE) models of developed structure specimens.

**Figure 21 polymers-12-02120-f021:**
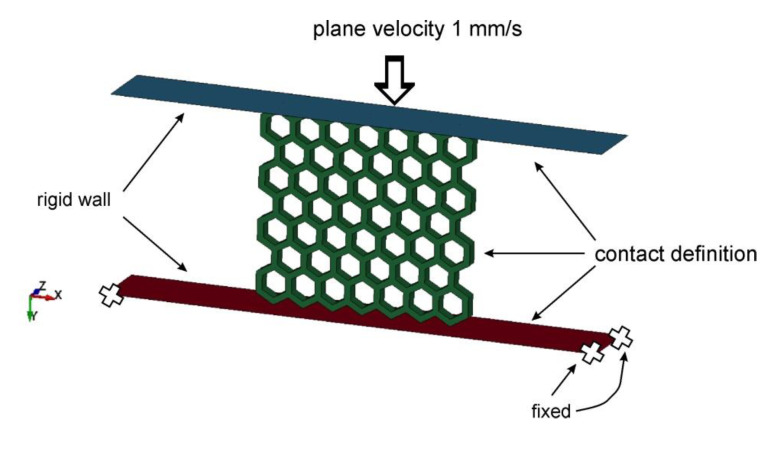
The main boundary conditions adopted for the numerical simulations to reproduce the experimental tests.

**Figure 22 polymers-12-02120-f022:**
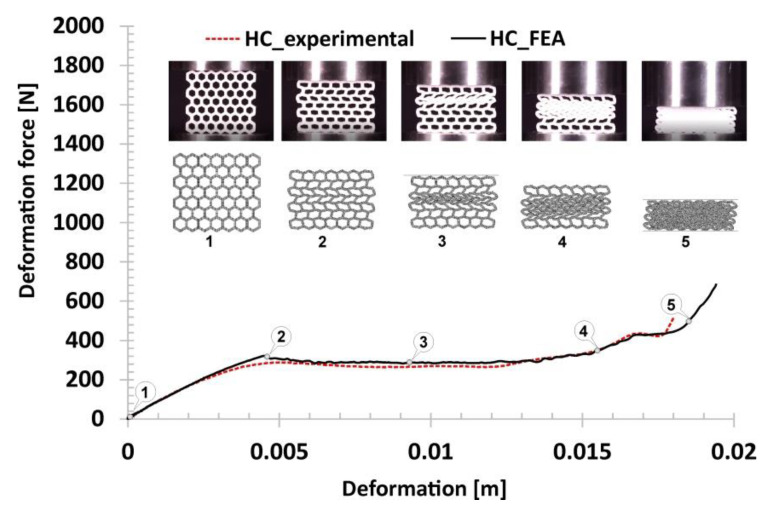
The results of the honeycomb compression tests obtained based on the FE analysis.

**Figure 23 polymers-12-02120-f023:**
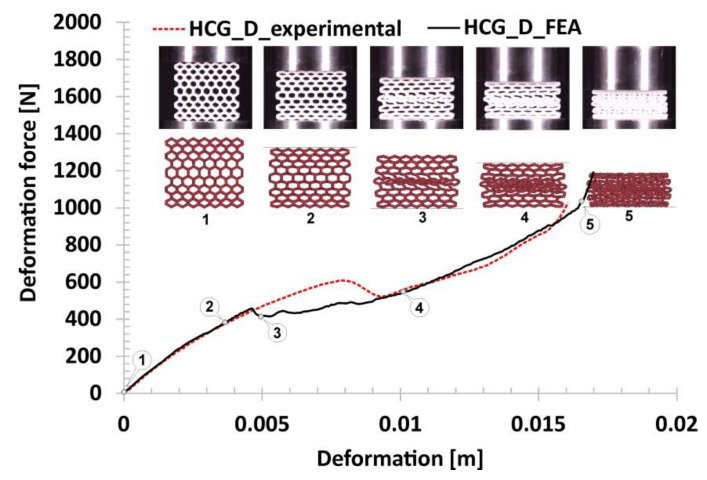
The results of the gradually decreasing honeycomb compression tests obtained based on the FE analysis.

**Figure 24 polymers-12-02120-f024:**
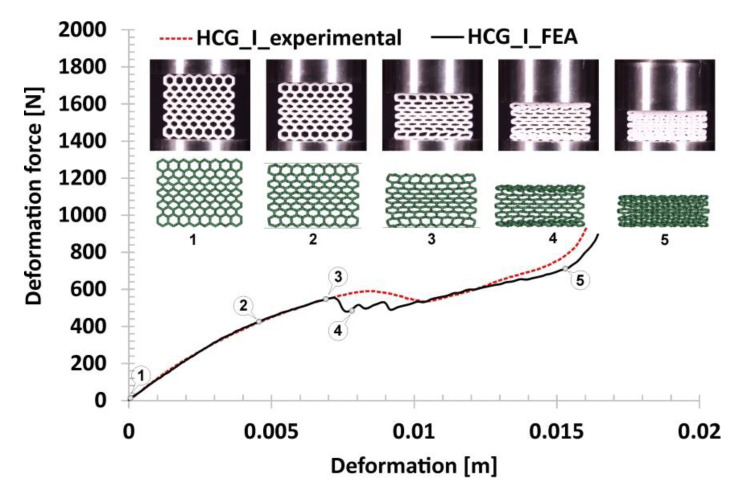
The results of gradually increasing honeycomb compression tests obtained based on the FE analysis.

**Figure 25 polymers-12-02120-f025:**
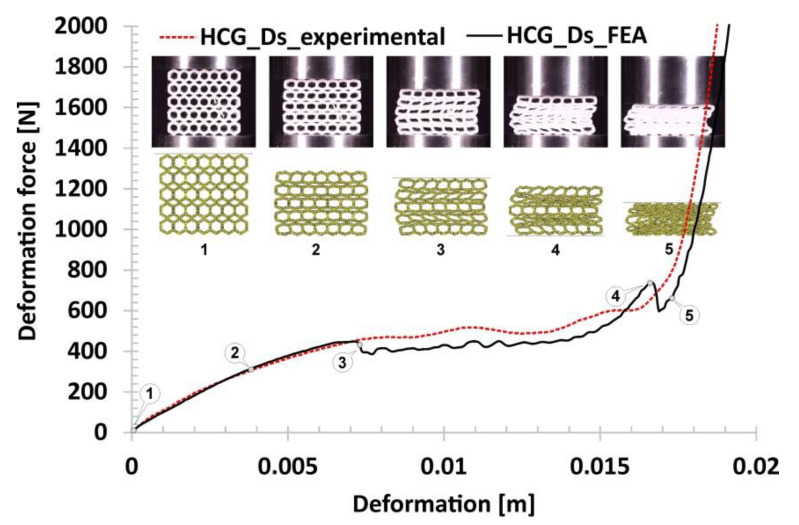
The results of discrete gradient honeycomb compression tests obtained based on the FE analysis.

**Figure 26 polymers-12-02120-f026:**
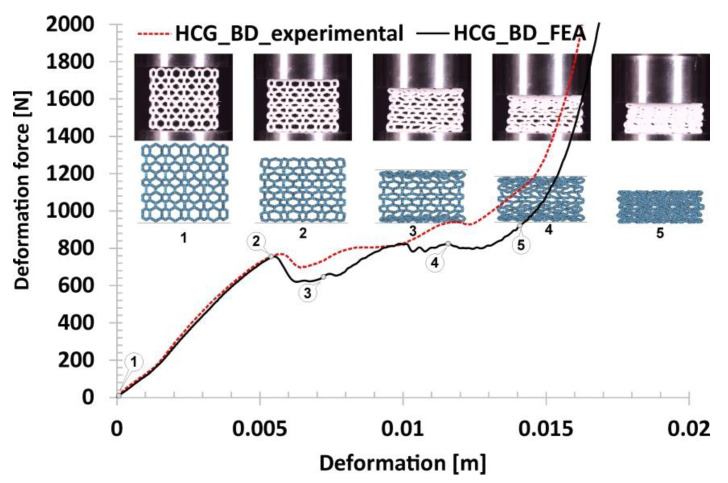
The results of bidirectional gradient honeycomb compression tests obtained based on the FE analysis.

**Figure 27 polymers-12-02120-f027:**
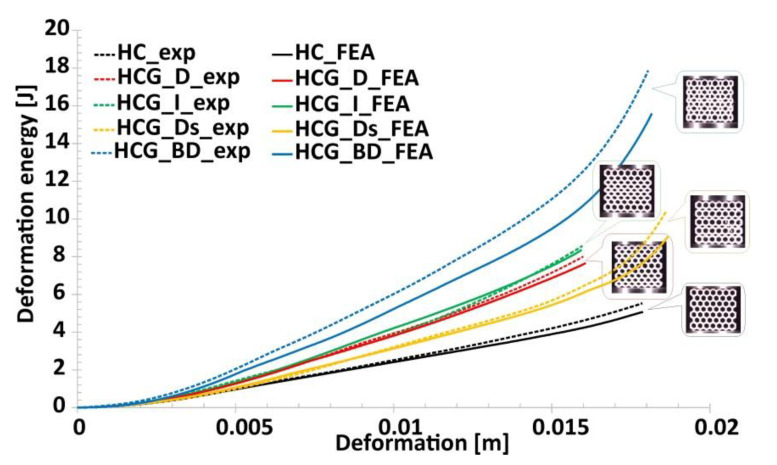
Comparison of deformation energy plots obtained from the FE analysis and experimental tests.

**Table 1 polymers-12-02120-t001:** Geometrical specifications of developed lattice structures.

No.	Wall Thickness [mm]	Dimensions of Specimen [mm]	Theoretical Relative Density *ρ*_rel_ [–]
Specimen No.1 (HC)	1.0	31.4 × 34.4 × 20.0	0.37
Specimen No.2 (HCG_D)	1.0	31.4 × 34.4 × 20.0	0.39
Specimen No.3 (HCG_I)	1.0	31.4 × 34.4 × 20.0	0.39
Specimen No.4 (HCG_Ds)	1.0	31.4 × 34.4 × 20.0	0.38
Specimen No.5 (HCG_BD)	1.0	31.4 × 34.4 × 20.0	0.42

**Table 2 polymers-12-02120-t002:** Mechanical properties of TPU 95 Polyflex material available on the producer website.

Mechanical properties	Density (ASTM D792)	Melt Index (210 °C, 1.2 kg)	Elastic Modulus (X-Y) ASTM D638	Tensile Strength (X-Y) ASTM D638	Elongation at Break (X-Y) ASTM D638	Shore Hardness ASTM D2240
TPU 95-Polyflex	1.20–1.24	3–6 (g/10 min)	9.4 ± 0.3 (MPa)	29.0 ± 2.8 (MPa)	330.1 ± 14.9 (%)	95 A

**Table 3 polymers-12-02120-t003:** Technological tests of 3D printing parameters (red dotted line—selected group of 3D printing parameters).

Parameters Group	Nozzle Temperature (°C)	Bed Temperature (°C)	Wall Printing Speed (mm/s)	Infill Printing Speed (mm/s)	Layer Height (mm)	Line Width (mm)	Flow Ratio (%)
Parameters set No.1	215.0 ± 0.2	60.0 ± 0.1	6.00	12.00	0.2	0.4	145.0
Parameters set No.2	215.0 ± 0.2	60.0 ± 0.1	6.00	12.00	0.2	0.4	125.0
Parameters set No.3	215.0 ± 0.2	60.0 ± 0.1	12.00	12.00	0.2	0.4	100.0

**Table 4 polymers-12-02120-t004:** Results of wall thickness measurements depending on applied 3D printing parameters set (green line—nominal dimensions, red lines – estimated values of wall thickness).

Set Groups	W1 [mm]	W2 [mm]	W3 [mm]	W4 [mm]	S1 [mm]	S2 [mm]	S3 [mm]	S4 [mm]	H1 [mm]	H2 [mm]	H3 [mm]	H4 [mm]
**Nominal Values**	0.60	0.80	1.00	1.2	0.60	0.80	1.00	1.2	0.60	0.80	1.00	1.2
Set No.1	0.89 ± 0.02	1.06 ± 0.02	1.13 ± 0.03	1.32 ± 0.03	0.94 ± 0.05	1.12 ± 0.06	1.27 ± 0.04	1.48 ± 0.09	0.94 ± 0.04	1.11 ± 0.03	1.28 ± 0.03	1.50 ± 0.05
Set No.2	0.80 ± 0.02	1.01 ± 0.02	1.13 ± 0.03	1.33 ± 0.03	0.75 ± 0.04	0.94 ± 0.04	1.21 ± 0.05	1.39 ± 0.06	0.73 ± 0.05	0.93 ± 0.06	1.11 ± 0.09	1.32 ± 0.09
Set No.3	0.72 ± 0.02	0.84 ± 0.02	1.28 ± 0.03	1.36 ± 0.03	0.66 ± 0.02	0.84 ± 0.03	1.13 ± 0.03	1.31 ± 0.04	0.67 ± 0.02	0.84 ± 0.03	1.11 ± 0.03	1.38 ± 0.04

**Table 5 polymers-12-02120-t005:** Adopted TPU 95 Polyflex stress–strain characteristics.

**Strain [-]**	−0.83	−0.78	−0.71	−0.69	−0.64	−0.43	−0.33	−0.23	−0.14	−0.09	−0.04	−0.03
**Stress [MPa]**	−270.48	−156.06	−83.28	−69.07	−50.23	−20.11	−14.95	−10.02	−6.02	−3.91	−1.96	−1.05
**Strain [-]**	0.0	0.04	0.1	0.23	0.71	1.45	2.34	2.67	2.96	3.24	3.55	3.81
**Stress [MPa]**	0.0	2.0	4.01	5.99	7.99	10.01	14.99	20.02	25.0	29.97	34.99	38.0

**Table 6 polymers-12-02120-t006:** Software parameters applied in Ls Dyna to describe the Simplified Rubber Material (SRM) model.

RO Density	KM Linear Bulk Modulus	SGL Specimen Gauge Length	SW Specimen Width	ST Specimen Thickness
1100.00 kg/m^3^	1.650 × 10^9^ Pa	1.00	1.00	1.00

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
