# Peer review of "Deformation Process of 3D Printed Structures Made from Flexible Material with Different Values of Relative Density"

_polymers, 2020, doi:10.3390/polym12092120_

Round 1

Reviewer 1 Report

The paper clearly describes the characteristics of original 3D printed cell promizing materials.

The description of the process and the materials themselves is very well explained lines 136-228. The quality control of materials is presented in an original way up to line 276. These two parts are very positive elements of the publication.

Then, the experimental process is well described, with tensile and compression tests. For instance, the Fig.14 to 17 shows clearly the non-linearity of the cellular structure, and also the effects of local buckling and crushing.

The hyperelastic proposal, with elementary numerical models, consistent with the measures. The model configuration presented Fig.21 is realistic, and the implentation on LS-Dyna with low velocity (1 mm/s) and Ogden formulation for hyperelasticity is adapted. And the obtained results are coherent qualitatively and in magnitude.

The bibliography is convenient.

The paper is interesting, but could be improved with the small modifications described hereunder :

  • The strong point is the experimental analysis. In paragraph 6 (which is essential) it is necessary to emphasize on the representation of the physical phenomena such as : local buckling, material saturation, crushing. This could be made by one or two additional small paragraphs (2-3 lines each).
  • The conclusion is a little short ; please precise the dynamic (and even thermodynamic) perspectives.

With these two light improvements, the paper can be published.

Author Response

Dear Reviewer, 

In the beginning, the Authors would like to express their great thanks for the effort in a very thorough analysis of the article in terms of its scientific aspects. The presented comments are very valuable and helpful to prepare the revised version of the manuscript. The comments have been deeply studied and final corrections have been introduced.

Particular comments 

Comment 1)

In paragraph 6 (which is essential) it is necessary to emphasize on the representation of the physical phenomena such as local buckling, material saturation, crushing. This could be made by one or two additional small paragraphs (2-3 lines each).

Response:

Thank you very much for your valuable comment. Mentioned information is crucial not only in terms of computer simulations but also from an experimental point of view. Taking into account the Reviewer’s remark, the authors decided to modify paragraph No.4. Based on the deformation force plots presented in Figures from 13 to 17, the main stages of the deformation process were discussed. Furthermore, the identification and description of the main mechanisms responsible for the deformation of the structures have been added. The applied TPU 95 material demonstrates high flexibility which results in an initiation process of buckling and bending mechanisms while the deformation process. Specimens subjected to testing after the force unloading come back to the initial shape. No shearing or cracking effects are detected. Nevertheless, proposed variants of specimen topologies result in a different way of deformation.

Comment 2)

The conclusion is a little short; please precise the dynamic (and even thermodynamic) perspectives.

Response:

Thank you very much for your valuable comment. Taking into consideration the planned range of structure application it is advisable to continue these investigations with consideration of the dynamic studies. They will consist of three main parts: characterization of material behaviour under dynamic loading conditions, dynamic compression tests, as well as computer simulations. Compression tests will be carried out with the use of a dedicated version of Split Hopkinson Pressure Bar with 40.0 mm diameter in a direct impact configuration. The performed characterization of the material mechanical properties will allow for the definition of TPU 95 Polyflex strain rate sensitivity. This feature will be taken into account in further numerical studies. The applied initial and boundary conditions in computer simulations will reflect the conditions of experimental tests. Based on the successfully validated numerical model and applied the constitutive relation of the material it will be possible to start the optimization process.

Reviewer 2 Report

The manuscript studied the deformation of several 3d printed structures through FFF method and based on TPU. Since the 3d shapes of FFF printed objects could be very different from objects made from compression molding and casting, which usually produce solid films, the mechanical properties of FFF printed polymer objects are more complicated, e.g., if an object with vasculature structures is deformed, not only the intrinsic stress-strain relationship of the material itself should be considered, the structure itself would also contribute to the deformation. Luckily, due to the precision of FFF printing technique, such structures are precisely controlled and thus could be simulated by mathematical methods. Therefore, it is the topic of this manuscript is meaningful.

Commercialized substrate TPU was used in this manuscript, which makes the study simpler and more focused on the deformation. The background and methods are well described. From the results, especially the compression tests of honeycomb structures with different topologies, the discrete honeycomb is not as stable as the others. For the other three types of honeycomb, the authors studied five stages of each, among which the trend from stage 2 to stage 3 and from stage 3 to stage 4 are the most differentiated transitions. It is recommended to highlight and expand the transitions in between stage 2 and stage 4.

Also, it could be observed that simulated deformation curves show similar trends but are smoother. It is understandable that realistic tests show more fluctuation since material fabrication could introduces in inevitable flaws.

As a summary, the manuscript is competitive and should be published.

Author Response

Dear Reviewer, 

In the beginning, the Authors would like to express their great thanks for the effort in a very thorough analysis of the article in terms of its scientific aspects. The presented comments are very valuable and helpful to prepare the revised version of the manuscript. The comments have been deeply studied and final corrections have been introduced.

Particular comments 

Comment 1)

The background and methods are well described. From the results, especially the compression tests of honeycomb structures with different topologies, the discrete honeycomb is not as stable as the others. For the other three types of honeycomb, the authors studied five stages of each, among which the trend from stage 2 to stage 3 and from stage 3 to stage 4 are the most differentiated transitions. It is recommended to highlight and expand the transitions in between stage 2 and stage 4.

Response:

Thank you very much for your valuable remark. The mentioned section was modified in accordance with the Reviewer's suggestion. Furthermore, Figure 16 and Figure 17 were modified in a way that should better present the differences in the deformation process.

Reviewer 3 Report

I commend the authors for their research on TPU material used in 3D printing and its evaluation based on cell geometry. Introduction was through, all required elements of the article were enough described. Materials and methods were brief. Methodology followed was straight and to the point. All the images, tables and graphs were well annotated and clearly presented. Conclusions drawn from the results were logical.

It would have been great it 'statistical significance' sub-section was added to the manuscript. Essential details including number of samples evaluated per test, type of statistical analysis performed for the mechanical properties evaluation were lacking. Similarly, Table 3 and Table 4 do not have +/- SD values. Please mention n number and +/- SD. I would recommend publishing after these minor edits

Author Response

In the beginning, the Authors would like to express their great thanks for the effort in a very thorough analysis of the article in terms of its scientific aspects. The presented comments are very valuable and helpful to prepare the revised version of the manuscript. The comments have been deeply studied and final corrections have been introduced.

Comment 1)

It would have been great it 'statistical significance' sub-section was added to the manuscript. Essential details including number of samples evaluated per test, type of statistical analysis performed for the mechanical properties evaluation were lacking.

Thank you very much for your valuable remark. It’s true that statistical analysis should be included in the manuscript. However, due to the high repeatability of obtained results of experimental tests under quasi-static loading conditions, the authors decided that three attempts were satisfactory. These remarks will be included in further experimental studies carried out under dynamic loading conditions. Based on current results the standard deviation analysis was carried out to define the deviation of TPU mechanical properties (maximum value of uniaxial tensile stress as well as uniaxial tensile strain ). Furthermore, the standard deviation value of the representative honeycomb structure wall thickness after the manufacturing process was estimated.   

Comment 2)

Similarly, Table 3 and Table 4 do not have +/- SD values. Please mention n number and +/- SD. I would recommend publishing after these minor edits

Thank you very much for your suggestion. The values of standard deviations were added to Table 3 and Table 4. However, in the case of Table 3, only for these parameters which are generally available (temperature of the nozzle and table bed). Unfortunately, the authors did not find information regarding: - wall printing speed, infilling printing speed, layer height, line width as well as flow ratio. Nevertheless, based on carried out wall thickness measurements of the benchmark model, the standard deviation values were estimated and included in Table 4.